# Plants and Other Materials Used for Dyeing in the Present Territory of Poland, Belarus and Ukraine according to Rostafiński’s Questionnaire from 1883

**DOI:** 10.3390/plants12071482

**Published:** 2023-03-28

**Authors:** Piotr Köhler, Aleksandra Bystry, Łukasz Łuczaj

**Affiliations:** 1Faculty of Biology, Institute of Botany, Jagiellonian University, ul. Gronostajowa 3, 30-387 Kraków, Poland; piotr.kohler@uj.edu.pl; 2Dzikie Barwy, ul. Pomorska 98 lokal 108, 91-402 Łódź, Poland; kontakt@dzikiebarwy.com; 3Faculty of Biology, Institute of Biology and Biotechnology, University of Rzeszów, ul. Pigonia 1, 36-100 Rzeszów, Poland

**Keywords:** ethnobotany, natural dyes, traditional ecological knowledge, textiles, wool, flax, Easter eggs

## Abstract

Background: Traditional dyeing methods are practically forgotten in Poland. Józef Rostafiński included questions on the use of dyes in his ethnobotanical survey from 1883. Methods: 126 questionnaires contained information on dye plants. They were identified by the respondents using folk names or sometimes even Latin names. Folk names were analyzed by comparison with other literature. Several voucher specimens were also present. Results: 74 plant taxa were identified to genus or species level. The most commonly used were: onion (*Allium cepa*), brazilwood (*Caesalpinia brasiliensis* or *Paubrasilia echinata*), winter corn (mainly rye *Secale cereale)*, black alder (*Alnus glutinosa*), safflower (*Carthamus tinctorius*), apple (*Malus domestica*), birch (*Betula pendula*), oak (*Quercus robur*), and violet flowering spring flowers (mainly *Hepatica nobilis* and *Pulsatilla* spp.). Conclusions: Most species are well known in the literature about plant dyeing, but the paper provides extra details on the picture of dyeing traditions in Eastern Europe.

## 1. Introduction

Plants have been a source of dyes since the dawn of humanity, used to decorate human bodies, textiles, containers, and for artistic and religious painting, etc. [1,2,3,4]. Species used for traditional dyeing and the techniques employed have been recorded in some areas of the world [5,6,7,8,9,10,11,12,13,14].

Before the popularization of synthetic dyes in the 19th and 20th centuries, the art of dyeing was an important craft [15,16,17,18,19,20,21,22,23]. Apart from dyes of plant origin, a local species of insect, *Porphyrophora polonica* (Linnaeus, 1758), sometimes called the Polish cochineal, was used to make red paint. The larvae of this scale insect live on the roots of various herbs, especially those of the perennial knawel *Scleranthus perennis* L., which is common on the sandy soils of Central Europe. Before aniline, alizarin, and other synthetic dyes were invented, the insect had been of great economic importance. It was exported to other parts of Europe, but its gathering and red dye production collapsed after the discovery of America and the introduction of cochineal red from another insect species [1,4,15,23]. A few interesting publications on dyeing plants and techniques were published in 18th-century Poland and later in the former Polish territories occupied by Russia during the partitions (1772–1918), which are now part of Poland, Lithuania, Belarus, and Ukraine [24,25,26,27,28,29,30,31]. For example, Krzysztof Kluk (1739–1796), an eminent Polish naturalist, encouraged the cultivation and use of dye plants in his textbook [29]. In addition to basic and generally known plants, such as reseda (*Reseda lutea*), woad (*Isatis tinctoria*), and madder (*Rubia tinctoria*), he paid attention to a number of field and forest plants with dyeing properties. He enumerated the local raw materials used by the villagers, such as barberry bark and twigs, apple and alder bark, bowls of acorns, and many others [29]. In the 19th century, information on traditional dyeing methods and materials could be found in Józef Gerald-Wyżycki’s (1792–1868) herbal [30] and in Anna Ciundziewicka’s (1803–1850) *Gospodyni Litewska* [31]. In the 20th century, several more plant dye textbooks and monographs on plant dyeing materials were published in Poland [32,33,34,35]. The contributions of the Polish ethnologist Kazimierz Moszyński (1887–1959) in his *Kultura ludowa Słowian* (*Folk Culture of Slavs*) [22] and the Polish historian Elżbieta Kowecka (1929–2001) are especially important [15]. An interesting monograph on dyeing was recently published by the botanist Adam Kapler [36].

However, due to the large availability of cheaper industrial dyes (usually of synthetic origin), the whole tradition of natural dyeing is disappearing, being either preserved only among some oldest craftsmen or becoming completely obsolete. In the case of Poland, the latter is true. Apart from the use of onions to color Easter eggs, natural dyeing is completely forgotten and does not occur even in 20th-century ethnographic publications, apart from data from the Polish Ethnographic Atlas, mainly from 1983–1990, where, analogously to Rostafiński’s questionnaire, two questions were included [37]. One question concerned the kinds of bark used in dyeing, and the other was about Easter eggs [37]. The 19th-century ethnographic materials are also silent about this type of plant use, as reflected by the fact that Adam Fischer’s ethnobotanical dictionary, which contains a synthesis of Polish data on the folk use of plants, does not mention them [38].

A valuable contribution to documenting the forgotten traditional dye plants is a questionnaire published in 1883 by the botanist Józef Rostafiński, professor of Jagiellonian University in Kraków [39]. He issued it in 60 editions of various periodicals in the territories of the former Kingdom of Poland (Poland was divided into Russia, Prussia, and Austro-Hungary at that time). Section VIII contained three questions concerning dyes:

VIII. Dyes

(59) Do simple people dye flax or cannabis textiles or wool or leather themselves? What plants are used and what colors do they give? I would be grateful for specimens.

(60) With what do they dye Easter eggs?

(61) Do people still gather “czerwiec polski” (Polish cochineal) and from under what plants?

Some information on dyeing was also found scattered in answers to other questions, especially question no. 46 about *Carthamus tinctorius* L. (“Do people know the name krokosz and is this herb used for?”).

Rostafiński began his research career as a taxonomist. However, at the beginning of the 1880s, he became interested in the history and names of cultivated plants. In 1883, Rostafiński started his largest project connected with plant names. His concept was to collect Polish plant names and write a history of plant cultivation and use in the areas of the former Polish-Lithuanian Commonwealth.

Rostafiński’s work was only partly analyzed and published. The answers to some questions, e.g., concerning wild greens or fungi, were analyzed in full detail, while others are still waiting to be properly elaborated. Basic information on the historical background of the questionnaire can be found in the works of Köhler [39,40,41]. The replies inspired Rostafiński to alter the scope of the questionnaire twice; however, the questions about dye plants are present in all of them. The questionnaire was published in 1883 in a few dozen Polish-language periodicals in what was then the Russian Empire, Prussia, and Austro-Hungary, as between 1795 and 1918, Poland did not exist as an independent country and was partitioned between these three empires (see Figure 1 and Figure 2 for the geographical scope of answers).

The aim of the present study was to analyze Rostafiński’s questionnaire. We hypothesized that most of the traditional dyes recorded in the study are already known from other specialist literature. However, we hope to find at least some novel species that could be utilized.

Researching plant dyes is important not only from the point of view of recording traditional knowledge. The information is also of value to modern enthusiasts of plant dyes. In the last two decades, there has been an immensely increasing trend in using natural products such as wild vegetables [42,43], medicinal herbs and teas [44], and plant dyes. Several handbooks on traditional dyeing have been published recently [45,46,47,48].

## 2. Results

The largest number of answers was given to question no. 60 (about Easter egg dyes), with 125 answers altogether and 425 use reports. A much smaller number of answers (62) was given to question no. 59 (dyes for textiles or leather), i.e., 176 use reports. Question no. 61 turned out to be a failure. Only six answers containing 14 use reports were given, mainly concerning other dyes, and not a single description of the contemporary use of *czerwiec* (Polish cochineal) was sent to Rostafiński. Thirteen use reports come from answers to other questions.

As many as 74 taxa of identified plants were recorded to species or genus level, and 13 taxa remained unidentified (Table 1), not counting a few materials of animal or human origin (e.g., dog feces and human urine). A few group categories were also distinguished (lichens, grasses, cereals, hay).

The most commonly used plants were onion (*Allium cepa*), brazilwood (*Caesalpinia brasiliensis* or *Paubrasilia echinata*), winter corn (mainly rye *Secale cereale)*, black alder (*Alnus glutinosa*), safflower (*Carthamus tinctorius*), apple (*Malus domestica*), birch (*Betula pendula*), oak (*Quercus robur*), and violet flowering spring flowers (mainly *Hepatica nobilis* and *Pulsatilla* spp.).

Hues of yellow, green, and red were the most common colors obtained from plants (Figure 2). Onion (*Allium cepa*) was the most widely used dye to obtain yellow and brownish colors, mainly on Easter eggs.

The second most commonly mentioned dye was Brazil wood, used mainly for dyeing textiles shades of red. We could not identify the exact species as the same names are used to refer to two related taxa, *Caesalpinia brasiliensis* and *Paubrasilia echinata*, in Europe (for more discussion, see footnote 4 in Table 1). 

Green blades of cereals were the third in the frequency of use as dyeing materials. Mainly winter corn was used, especially rye (*Secale cereale*), which is usually sown in autumn and easily obtainable during Easter for dyeing eggs green. Oats (*Avena sativa*) and wheat (*Triticum* spp.) were also used for this purpose.

Another commonly mentioned plant was black alder (*Alnus glutinosa*), a native common tree in the area. This is a very interesting plant dye as different parts of the plant (bark, fruits, leaves, roots) used with different mordants can give various shades of brown, black, and yellow. Widely used for textiles, wool, and yarn, it was also applied to Easter eggs. However, another common native tree used for dyeing was birch (mainly the most abundant *Betula pendula*), whose leaves were applied to give yellow color to textiles, wool, and probably Easter eggs as well. The third common native tree used in dyeing was oak (mainly the most abundant *Quercus robur*), whose bark was applied to give dark (black, brown) hues to wool and flax.

Apple (*Malus domestica*) is the commonest fruit tree in central-eastern Europe, and, as such, its leaves and bark were easily available materials, mainly for dyeing wool and probably also Easter eggs yellow.

Another commonly used material was safflower (*Carthamus tinctorius*), an annual plant that was specially cultivated for dyeing purposes (for wool and Easter eggs) due to the attractive shades of yellow and pink achieved.

## 3. Discussion

### 3.1. Comparison with Other Studies

As shown in Table 1, most of the listed taxa are widely known in the dyeing industry or have been recorded by other studies. Nevertheless, this study is a valuable and large-scale documentation of dyeing practices in the Eastern European countryside. The prominence of a few well-known materials is visible. The studies of the Polish Ethnographic Atlas list only 19 species of dye plants, compared to 74 (plus unidentified taxa) in Rostafiński’s questionnaire from over a century before. This well illustrates the decrease in dyeing traditions between the 19th and 20th centuries.

There are very few records of the cultivation of *Rubia* in Poland, which is consistent with the observations of other authors that this plant was used in the dyeing industry but was not part of the Polish dyeing tradition [15,22].

The low presence of blue dyes (Figure 2) is no surprise. This was a sought-after color, but very few plants can provide it, in contrast to yellow, brown, or even red, often present in nature [21].

Rostafiński’s study confirmed that the tradition of using Polish cochineal red dye was dead by the end of the 19th c. No further ethnographic studies in Eastern Europe ever reported the use of this species as a dye.

No mushrooms were used as dyes in the 19th century, though the use of lichens was reported in some parts of Poland (Table 1). Unfortunately, there are no details or specimens that would enable their identification. Lichens have been used as a dye in some parts of the world, e.g., in the UK [51], and the information about their use in Poland is the only such record from Polish ethnographic literature. This can be counted as an achievement of Rostafiński’s study.

### 3.2. Identification Problems

Historical data usually do not have voucher specimens attached [59]. That is why some species were not identified either at the genus or species level. We faced this problem for most questionnaires, apart from Federowski’s [60], which included a detailed herbarium that also enabled the identification of taxa in other questionnaires. 

One of the problems was distinguishing *Carthamus tinctorius* and *Crocus sativus*. The former served as a cheap alternative to the latter. Another problem was distinguishing *Origanum vulgare* and *Thymus* spp. Both species can be called by folk names starting from *macier-*, *mater-*, meaning mother. Usually, when the red dye is concerned, we are dealing with *Origanum vulgare* [22], but possible identification problems may occur. We also had a problem distinguishing *Origanum vulgare* and *Chenopodium*, which tend to have similar names (lebiodka, lebioda). We were also unsure which species of Caesalpinieae was used for red paint (see Table 1). Another issue is distinguishing *Hepatica nobilis* and *Pulsatilla* spp. Both of these taxa were probably used.

### 3.3. Weeds and Woody Species as Dyeing Plants

It must be noted that many dye plants were “weeds” in pastures, e.g., *Origanum vulgare*, *Arctostaphylos uva-ursi,* or woodland plants of little fodder value, e.g., *Lycopodium* spp., *Hepatica nobilis*, *Rhamnus cathartica*, *Berberis vulgaris*, *Prunus spinosa*. Others were cereal weeds (*Centaurea cyanus*, *Papaver rhoeas*). Thus, the harvesting of the abovementioned species came with an extra benefit for farming by removing weedy or inedible species. 

Tree leaves, bark, and roots are another important category of dyeing plants. Out of the 10 most commonly used dyeing ingredients, three taxa were native common tree species: alder, birch, and oak; apple was the most common fruit tree. Given their abundance, they could easily be utilized as a dye. Trees in gardens and small woods often have their lower branches chopped off to increase the growth of the main stem, so the parts used for making dyes could have been just farming by-products.

### 3.4. Easter Eggs

This study is an important contribution to the issue of dyeing Easter eggs. Dyeing Easter eggs has been a custom widespread in Eurasia since the early ages of Christianity. Depending on the local tradition, the eggs could be later eaten, or only empty shells were used for dyeing. Examples of ritual egg dyeing are also known from pre-Christian times [61,62]. The decorations of Easter eggs were a subject of ethnographic research in Poland, but mainly in the context of their patterns and customs associated with them, not the species of plants used for dyeing [62,63], apart from the study of the Polish Ethnographic Atlas [37]. Recording the use of 53 taxa for dyeing Easter eggs may help in preserving this tradition. This number is quite impressive, considering that, for example, Guarrera recorded only three species used from a few regions of Italy [64], and only 13 taxa were recorded by the studies of the Polish Ethnographic Atlas for dyeing eggs in Poland [37].

## 4. Materials and Methods

We extracted the information concerning the researched questions from a database of letters written to Rostafiński in response to his questionnaire, which was created by the first author (P.K.). Additionally, we included two published works of imminent Polish ethnographers, Michał Federowski (1853–1923) [60] and Zygmunt Gloger (1845–1910) [65], which were structured using Rostafiński’s questionnaire and can be treated as the conceptual part of this project, i.e., responses to the questionnaire which were never sent to Rostafiński. Altogether, our database included 640 records from 126 respondents who provided meaningful information on plant dyes.

Plants were identified using standard methods applied in historical ethnobotany (compare the studies listed by da Silva et al. [66] and summarized by Lardos [67]), i.e., comparing available voucher specimens, folk names recorded in other sources, uses reported in previous publications and geographical distribution and abundance of the taxa used. The credibility of such historical identifications was also extensively discussed by Łuczaj [59]. A similar methodology has been used in other historical ethnobotany publications in the same special issue, *Historical Ethnobotany: Interpreting the Old Records* (e.g., [68,69,70]. Identification was facilitated by the voucher specimen collection supplied by Federowski. Twenty-six informants supplied 53 scientific names of plants, which seemed trustworthy in most cases (see the discussion on *Hepatica nobilis* and *Pulsatilla* in the Discussion). Common plants were also identified by comparing the folk names supplied with other sources on folk botany, e.g., Fischer’s dictionary [38].

When the provided local names of plants have been exclusively and commonly used for a certain genus or species throughout the study area, the scientific name of the genus or species was assigned to the local name. For example, “cebula” is the main name for *Allium cepa* L. used exclusively for this species. “Olcha” is used for the genus *Alnus*, “dąb” for oak (*Quercus* sp.), and no other plants have ever been called these names. Additionally, distribution maps of the taxa were checked to ensure they occurred in the studied localities, though most of the taxa used are very common species with large ranges. In the case of 13 plant names, trustworthy identification was impossible (Table 1).

## 5. Conclusions

Remnants of traditional plant dyeing knowledge were saved by Rostafiński in his 1883 study. His work revealed that the tradition of plant dyeing in Poland, Belarus, and Ukraine was already disappearing in the 19th century. The plants and other ingredients used to make dyes reported in this study are usually widely known species and ingredients used for dyeing. However, the data provided may be helpful in restoring traditional textile production and preserving the tradition of dying Easter eggs in Poland, Belarus, and Ukraine.

## Figures and Tables

**Figure 1 plants-12-01482-f001:**
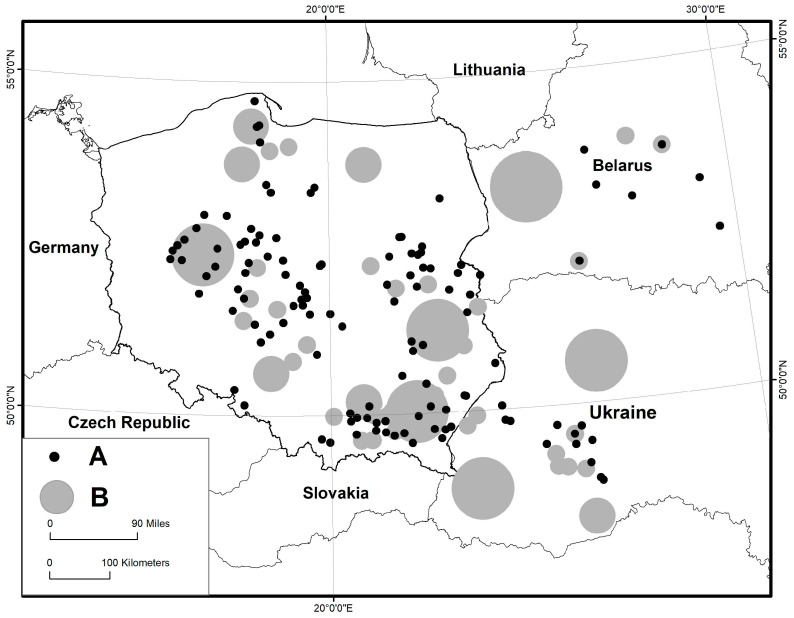
The distribution of records for dyes of organic origin in our study; A—places, B—regions.

**Figure 2 plants-12-01482-f002:**
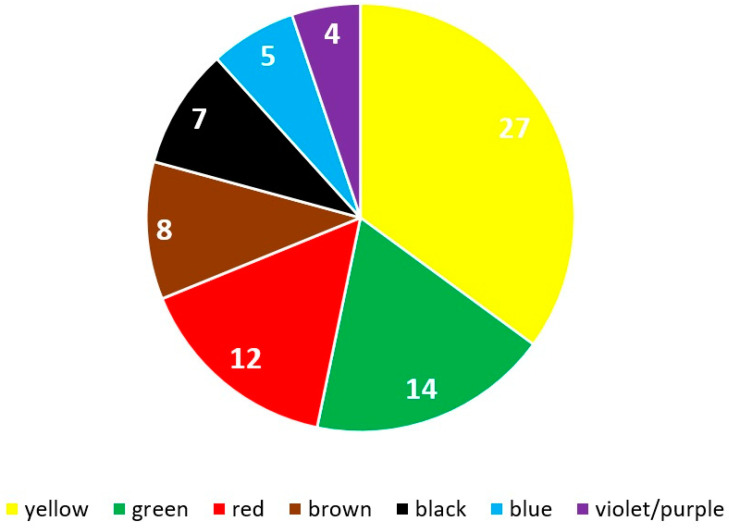
Number of species used for obtaining colors from plant dyes.

**Table 1 plants-12-01482-t001:** Natural dyes recorded by Rostafiński in 1883.

Scientific Name	Local Name	Part	UR	Use	Main Color	Geography	Use in Other Sources
*Alcea rosea* L. (identification uncertain)	malwa	fl, l	4	Easter eggs?	green (sapphire)		Used by amateur dyers for pink, blue, and gray colors [46]; roots used for eggs in Kostrzyń near Białobrzegi.
*Allium cepa* L.	cebula, cybula, dymka	outer layers of onion	150	Easter eggs, sheepskin coats	yellow (red-brown, brick-yellow)	throughout PL and BE	Mainly used to dye wool and eggs [37]. The longer the dyeing time, the redder the wool or eggshells. With a small amount of raw material and a short immersion time, the eggs or yarn turn yellow (the color of turmeric). With alum, the stained wool achieves more luminous colors—from golden yellow to brown saturated red [32].
*Allium* cf *schoenoprasum* L.	szczypiór	sh	1	Easter eggs	n. sp.		
*Alnus glutinosa* (L.) Gaertn.	olcha, olszyna (PL), aleszyna (BE)	bk, (r, c)	37	wool, flax yarn, Easter eggs; mordants: alum and sulfate, sometimes just iron filings, beetroot kvass	black, yellow, brown	throughout PL, BE, UA	A typical folk dye, alder was not used by manufactories or industry. Used to dye fibers (thread, wool), leather, and eggs [15,37]. Brown and beige on alum mordant (warm shades), and grey to brown and deep black on iron mordant. Such colors are given by the cones and bark, while leaves and catkins (inflorescences) give shades of yellow with alum and green-brown with iron ([32] and A.B.’s own experiments).
*Anethum graveolens* L.	koper	n. sp.	1	Easter eggs	yellow	Olszany (Przemyśl, PL)	
*Arctostaphylos uva-ursi* (L.) Spreng.	muczennik, muczennicznik, muczernik, mączennik	wh, r	5	textiles; dried, pounded to powder in a wooden mortar ^1^.	brown, black	Słonim area, Bobrujsk, Weleśnica, Jeziora (Grodno), Ihumeń (BE)	Yellow, gray and black [32,34].
autumn-sown cereals (winter corn)	ozimina, zboże	sh	17	Easter eggs	green		Widely used for eggs [37].
*Avena sativa* L.	owies	sh	2	Easter eggs	green (fresh shoots), yellow (straw)	PL: Staszkówka (Ciężkowice), Tarnów	
*Berberis vulgaris* L.	berberys	r, bk, br	9	Easter eggs? wool, flax	yellow	throughout (PL); Lipów (Rzeczyca county), Nieśwież, Łuck, Mińsk (BE)	Widely used around the world [46]. According to Kluk [29], peasants use barberry bark to dye yellow—he did not specify what. Used for brown dye in Kłudzie near Lipsko [37].
*Beta vulgaris* L.	burak		4				Probably only used for dyeing eggs. There are no sources about the use of beetroot for dyeing fibers. It is a pH-sensitive dye and gives unstable dyes (A. B.’s own experience).
*Betula* sp. (mainly *B. pendula* Roth)	brzoza (PL), bierioza (BE)	L (bd, bk)	12	Easter eggs? wool, flax, cotton; alum mordant mentioned in 1 answer	yellow, (green, yellow-green)	throughout (PL); Naliboki (Mińsk, BE); NW Belarus; Mikulińce, Tarnopol, Zbaraż, Skałat, Nowesioło (UA)	Widely used. Good for dyeing eggs, fabrics, and yarns. The leaves give yellow colors (leaves) with alum and greenish with iron. According to sources [32,34,37,45,46], birch bark dyes brown; A. B.’s experience shows that it also gives shades of pink and dirty pink, and pinkish red can be reached in an alkaline environment.
*Brassica oleracea* L. (probably var. capitata f. rubra)	kapusta	juice	2	flax textiles, Easter eggs?	n. sp.	Central Poland —lax textiles; Grąziowa nad Wiarem (SE PL)—probably Easter eggs	Doubtful for linen, as anthocyanin dyes do not bind permanently to cellulose fibers. Good for eggs, as its pH is sensitive and gives many shades, from pink to green (own experience, A.B.).
*Caesalpinia brasiliensis* L. or *Paubrasilia echinata* (Lam.) Gagnon, H.C.Lima and G.P.Lewis ^4^	brezylia, brazylia, bryzylia, farnebuk, farembak		73	Easter eggs	red (black, brown, dark blue, violet)	throughout	Imported. See footnote no. 4.
*Caltha palustris* L.	łotoć		1	Easter eggs	yellow		
*Cannabis sativa* L.	konopia	n. sp.	1	Easter eggs?	green	Pińsk (BE)	A.B. once used the hemp herb as a dye, and a greenish tint appeared on the wool fibers. Wool often reacts similarly to eggshells (due to protein content), so shades of green are possible on eggs.
*Carthamus tinctorius* L.	krokosz (mainly), also: krokost (PL), krokos (PL and BE), ćwitłuszka, ćwitłycia (UA)	n. sp.	30	Easter eggs? wool	yellow (red or pink in acid pH)	throughout (PL); SW UA	Safflower is still used in many countries [46] but is a difficult raw material, so in folk culture, it probably served only as a yellow dye for wool or eggs. According to Kluk [29], safflower was used to create varnishes and thick paints. Abroad, it was most often used to dye silk pink. By working with it for a long time and in stages, a bright pink or red color can be achieved from the flower stamens [46].
*Centaurea cyanus* L.	haber, modraki (PL); wołoszka, wałoszka (BE)	fl—for blue, l, r—for yellow or green	4	Easter eggs	blue, yellow, dirty green	Pleszew (Wielkopolska), Brus (Włodawa) (PL); Nieśwież, Słuck, Mińsk (BE), NW Belarus	Not mentioned in the literature, although there are modern recipes for dyeing with cornflower on the Internet.
*Cichorium intybus* L.	cykorya	n. sp.	2	Easter eggs?	n. sp.	NW Poland: Wielkopolska and Bielno (Świecie)	Leaves are used for dyeing in Iran [49].
*Coffea* sp.	kawa	n. sp.	2	Easter eggs?	n. sp.	Pomerania: Wysin (Kościerzyna), Kociewie	Gives beige and brown. Probably coffee dregs were used.
*Corylus avellana* L.	leszczyna	bk	1	textiles?	probably brown	Kaszuby (Pomerania)	Hazelnut bark with alum gives a saturated shade of yellow ([32] and A.B.’s experience); green-brown was also reported [37].
*Crocus sativus* L. (rarely) (or frequently *Carthamus tinctorius* L.)	szafran	n. sp.	8	Easter eggs	yellow	throughout (PL), Mościska, Jaworów, Lwów area (UA)	Saffron is a widely known, expensive source of yellow.
*Dianthus* sp.	goździki leśne	n. sp.	1	textiles?	yellow	Pińsk area (BE)	
*Canis familiaris* L.	pies (psie łajno)	feces	1	collected by boys for dyeing Morocco leathers (safiany)	red	SW Ukraine	
*Fagopyrum esculentum* Moench	hreczka	chaff	1	textiles?	yellow	Mikulińce, Tarnopol, Zbaraż, Skałat, Nowesioło (UA)	Works very well on cellulose fibers, turning them brown, beige, or yellow (A.B.).
*Fagus sylvatica* L.	buk	bk	1	textiles?	probably brown	Kaszuby (Pomerania)	Dyes brown, beige, and brown with a shade of pink very well (A.B.). Used to dye flax [37].
*Frangula alnus* Mill.	kruszyna	n. sp.	1	textiles?	blue	Lipów (Rzeczyca, BE)	Used to dye fabrics, wool, yarn, eggs (yellow, red, and green, depending on the mordant used) [37,46], and fruits, especially unripe ones. They give shades of green on copper mordant [32].
*Gallus gallus domesticus* (Linnaeus, 1758)	żółtko [chicken yolk]	yolk	1	sheepskin coats	red (one of the ingredients)		
*Genista tinctoria* L.	janowiec (PL), zinowat, zinówka, żołtucha (BE)	Fl	2	n. sp., probably textiles; alum mordant (1 answer)	yellow	Międzyleś, near Radzymin (PL); Weleśnica (BE)	Flowering shoots dye yellow. Widely used in Europe [32,46].
*Haematoxylum campechianum* L.	kempesz, kampesz	wd	4	Easter eggs?		PL: Biecz, Janów Lubelski, Mazury, Łyczkowice (Skierniewice)	Imported. Mainly used for textiles and yarn, cited by most dyeing handbooks [32,46]. Treated twice with alum Turns violet with alum, blue or grey with iron, and blue with sodium bicarbonate.
hay	siano	hay	3	Easter eggs	dark yellow	Zabłocie (Łask), Kalisz and Wieluń county, Wojnicz	
*Helianthus annuus* L.	słonecznik	n. sp.	1	textiles?	yellow	Siedlce	
*Hepatica nobilis* Schreb. & *Pulsatilla* sp. (probably both used but mainly the former)	kocanki, sasanki, podlaszczka (PL); son, przelaszczka, sonczyki, praleski, wiośnianki, pierwiosnki (BE)	fl	11	Easter eggs (mainly), textiles; the dye was made by the extraction of flowers with vodka	blue, violet	central-western Poland: Silnica (Radomsko), Lisice nad Nerem (Koło), Częstochowa county, Wieluń county, Bielice (Kutno) Ostrzeszów, Zabłocie (Łask); Nieśwież, Słuck, Mińsk, Bobrujsk (BE)	*Pulsatilla* has been used for amateur dyeing [50].
*Homo sapiens* Linnaeus, 1758	mocz [urine]	urine	1	as a preservative for blue dye in textiles ^3^	blue		Urine was commonly used as a mordant in folk dyeing, especially for building indigo vats of indigo or woad (fermentation vats with settling clarified urine) [15].
*Hordeum vulgare* L.	jęczmień	n. sp.	1	Easter eggs?	n. sp.	Grąziowa nad Wiarem (SE Poland)	
*Hypericum perforatum* L.	trojca świataja	fl	1	vodka	n. sp.	Przemyskie (PL); Czartkowskie i Tarnopolskie (UA)	Mostly used with oil and alcohol but also for dyeing textiles yellow [46].
*Juglans regia* L.	orzech włoski	fruit rind	2	Easter eggs	n. sp.	Jarosław, Cieszanów (PL)	A classic dye for wool that is also used for textiles [32,34,37,46]. Its surprising absence in the materials can be attributed to the small prevalence of walnuts in the 19th c. in the area.
*Juniperus communis* L.	jałowiec	fr	1	wool	n. sp.		The pseudofruits dye brown [32].
Lichens	mech z kamieni, wilcz	wh	6	wool	brown (olive, red—depending on the kind of lichen)	northern Poland (Kaszuby; Polskie Brzozie near Brodnica; Kociewie; Suszyn, Stępów, Wola Stępowska, all near Gostyń)	Lichens were widely used in NE Europe [51]. Reported as “mech” from Regnów near Rawa Maz. and Kocierzew near Łowicz [37].
*Linum usitatissimum* L.	len	n. sp.	1	Easter eggs?	n. sp.	Jeżewo near Borek	
*Lithospermum* cf *officinale* L.	wróble proson. sp.	roots	3	Easter eggs, wool, yarn, flax	red	Ozorków, Zgierz, Konstantynów, Pabianice, Tomaszów (PL)	Another species of the genus, *L. arvense*, was used as a dye in Poland [36].
*Lupinus* sp.? (uncertain identification by Rostafiński)	nogurje, zonagurje, nagwięć, zonagwięć	n. sp.	1	textiles	n. sp.	Niedźwiadka near Łuków	
*Lycopodium complanatum* L., and *L. clavatum* L. (and possibly other species from this genus)	zielenica, zeglen, swarzybaba	n. sp.	4	textile (flax, cotton), Easter eggs	green, yellow, red	Eastern Poland: Łuków, Sokołów Podlaski, Węgrów (esp. in Rażny), Włodawa (e.g., in Brus), Janów Lubelski (PL); Weleśnica (BE)	Used to dye wool [32]. In Iron Age Finland, it was boiled for several days to produce an alum mordant [52]. *Zielenica* was reported as used for eggs in Gnieczyna near Przeworsk and Zalesie near Siemiatycze [37].
*Malus domestica* (Suckow) Borkh.	jabłoń [mainly], also: jabłonnoka, kwaśnica	l, bk	19	wool, Easter eggs?	yellow, (green, red)	throughout (PL); Bobujsk (BE); Mościska, Jaworów, Mikulińce, Tarnopol, Zbaraż, Skałat, Nowesioło (UA)	Obtaining red from this species is difficult, but there are recipes from the early Middle Ages. In general, most sources on the practical side of dyeing [32,34,46] mention apple leaves as a raw material for dyeing yellow with alum and green with iron, and the bark as a raw material giving orange shades of yellow with glow or slightly pink colors for wool, as confirmed by A.B.’s experience.
*Origanum vulgare* L.	lebiodka, lebioda pusząca, lebioda (PL); macierduszka (BE)	flowering tops	8	textiles, yarn, wool	(dark) red	Eastern Poland: Międzyleś (Radzymin), Niedźwiadka and the whole Łuków area, Rażny near Sadowne (Węgrów); NW Belarus and Kuchcice (Ihumeń) (BE)	Widely used in north Slavic territories for red dye [22,29,31]. For example, Kluk writes, “village women boil it with alum to obtain red dye”.
*Papaver rhoeas* L.	polny mak, zajęczy mak, glapi mak	n. sp.	3	textiles	n. sp.	Pleszew area (western Poland)	Flowers formerly used to dye dark blue [29].
*Phaseolus vulgaris* L.? (with dark seeds)	fasola czarna	n. sp.	1	Easter eggs?	violet	Mikulińce, Tarnopol, Zbaraż, Skałat, Nowesioło (UA)	
Poaceae	trawa	sh	10	Easter eggs	green	throughout (PL)	
*Populus nigra* L. and P. x *canadensis* Moench	topola, jabrzędź	catkins	1	Easter eggs	green, (blue)	Maciejowice (Garwolin), Romanów (Włodawa)	Poplar catkins used to dye eggs brown in Starosiedlce near Iłża, Kostrzyń near Białobrzegi [37].
*Porphyrophora polonica* (Linnaeus, 1758)	czerwiec, czerwiec polski, maściki	insect body	4	textiles; e.g., flax and cannabis yarn (BE)	red	Podpniewki (Pniewy), Ostroróg (Szamotuły), Buk county, Kościelna Wieś (Kujawy) (PL); Ihumeń county (BE); Quote from Szamotuły (W. Zentkeler): they gathered worms called “maściki” and sold to apothecaries not only in Szamotuły county but also in Buk county	According to Jakubowski [23], use ceased before the 19th century. At that time, it was already an uneconomic raw material, as cochineal was imported instead.
*Potentilla erecta* L.?	termentyla	n. sp.	1	walking sticks, red stripes (together with alder phloem bitten in the mouth)	red	Jarosław, Cieszanów (SE PL)	
*Prunus domestica* L.	śliwa	bk	1	Easter eggs?	n. sp.	unknown location	Pink color [46].
*Prunus spinosa* L.	tarnina, tarń	bk, r	2	Easter eggs	black (roots), green (bark)	Grąziowa nad Wiarem—root, Rymanów—bark (SE Poland)	Brown color [32].
*Pterocarpus* sp.?	kraska, drzewo sandałowe	wd	1	Easter eggs	red	n. sp.	Imported. Formerly used for textiles and yarn [32,45,46], not to be confused with *Santalum.*
*Pyrus communis* s.l.	gruszka, przycierpka, lasówka	dried fr	1	leather	n. sp.	Rojówka near Tęgoborze (S Poland)	
*Quercus* spp. (mainly *Q. robur* L.)	dąb	bark	12	Easter eggs? wool, flax yarn, aprons, skirts, and corsets; with iron salts as	black, (brown, reddish)	throughout (PL); Ihumeń county, Lipów, Rzeczyca county (BE); Czortkowskie i Tarnopolskie (UA)	Was used for dyeing fabrics, wool yarn, and mordanting, as well as wood, hair, and fishing nets [37]. Produces shades of brown and beige with alum mordant and shades of gray and black with the addition of iron/iron sulfate in a different form. Black is the easiest to achieve on wool [32].
*Rhamnus cathartica* L.	sakłak, szakłak, szkłak	fr	4	flax, cotton, wool	brown, yellow, dark green, (black ^2^).	PL: Młyny (Strzelno), Sokołów, Węgrów, Włodawa, Janów Lubelski; UA: between Lwów and Żółkiew	Unripe buckthorn berries were used to dye fibers green instead of the in-house method (first dyeing blue, then yellow) [15]. *R. cathartica* and *R. frangula* were probably confused and used interchangeably.
*Rubia* cf *tinctorum* L., though some *Galium* sp. cannot be excluded	marzanna, mazonna	r	3	textiles, Easter eggs, cultivated	n. sp. [probably red]	Łódź, Pabianice, Ozorków, Janów Lubelski, also: Lithuania and Samogitia	People used madder very rarely. It was a typical plant of the dyeing industry [15], usually used to give red, pink, orange, brick, and brown colors. It is the most durable of the natural reds of plant origin, with no equal (e.g., brazilwood is photosensitive; all practical sources say that it is used for dyeing evening clothes rather than day clothes) [32,34,45,46].
*Rumex* sp.	szczaw dziki	n. sp.	1	textiles	yellow	Mikulińce, Tarnopol, Zbaraż, Skałat, Nowesioło (UA)	Used for wool [32]. A.B.’s experience suggests a bright yellow color with alum mordant.
*Salix* sp.	wierzba	bk from young twigs	1	Easter eggs		Tarnów county (Dębica, Dąbrowa, Żabno, Ropczyce)	Gives a light green color [37].
*Salvia officinalis* L.	szałwija	n. sp.	1	Easter eggs	n. sp.	Malbork (PL)	Used for dyeing rugs in Turkey [53].
*Sambucus nigra* L.	bez (PL), baznyk (UA)	fr, (bk)	4	Easter eggs	violet, dark	Jarosław, Cieszanów; Brus (Włodawa) (PL); Winniki (Sambor, UA)	Widely known, e.g., [32,34,46].
*Saponaria officinalis* L.	kukułyca	n. sp.	1	washing clothes before dyeing		Weleśnica (BE)	
*Secale cereale* L.	żyto	sh	39	Easter eggs	green	throughout	Easter eggs [37,54].
*Serratula tinctoria* L.	żółkwiło	n. sp.	1	wool	yellow	SE part of Biłgoraj area (E Poland)	Good yellow dye for fibers [32,34]
*Solanum tuberosum* L.	ziemniak	peel from tubers	1	Easter eggs?	n. sp.	Kartuzy, Kaszuby (PL)	
*Sorbus aucuparia* L.	jarzębina	bk	1	Easter eggs?	dark yellow	Wołyń (PL)	Bark is a good olive-colored dye for fibers [34].
*Spinacia oleracea* L.	szpinak	n. sp.	1	yarn	n. sp.	Tykocin, Zambrów (NE PL)	
*Tagetes* sp.	kupczaki pełne	n. sp.	1	wool	red	Iwanków near Borszczów (UA)	Depending on the mordant used, it dyes olive, yellow, brown, or orange. A good dye for both plant and animal fibers [34].
*Thymus pulegioides* L., *T. serpyllum* L. or *Origanum vulgare* L.	macierzanka (PL and BE), materynka (PL), matyrynka (UA)	n. sp.	3	wool	dark brown	Brus (Włodawa) (PL); Lipów (Rzeczyca) (BE); Czortków area (UA)	The name can be applied to both genera, though *Origanum vulgare* is most likely.
*Thymus pulegioides* L.	czabor, cząber	n. sp.	1	Easter eggs	green	NW Belarus	Identification confirmed by voucher specimen. *T. serpyllum* L. was probably used in a similar fashion for
*Triticum* sp.	pszenica	sh	6	Easter eggs	green	throughout (PL)	eggs [37].
*Urtica dioica* L. and *U. urens* L.	pokrzywa (PL and BE), krapiwa piekuszcza (BE, for U. dioica), rzeszka (BE for U. urens)	rt	3	Easter eggs	bright yellow	NW Belarus, esp. Jeziora near Grodno; SE part of the Biłgoraj area (PL)	Dyes bright green or yellow. Best used with mordants [46].
*Vaccinium myrtillus* L.	czarne jagody (PL), czernice (BE)	fr (dried or juice)	4	flax and cannabis textiles, e.g., kerchiefs	black	Kalisz area, Kcynia (Szubin) (PL); Naliboki (Pińsk) and Weleśnica (Mińsk) (BE)	The color is sensitive to light and pH change; most suitable for wool (A.B.’s own observations). Mentioned by [54] as a source of blue dye.
*Vaccinium* sp.	[only Latin name given]	fr	1	dried fruit in vodka for wool and flax; alum mordant	black		
*Vaccinium uliginosum* L.	maczało	n. sp.	1	threads	n. sp.		
*Viburnum opulus* L.	kalina	fr	1	decocted	red	Winniki (Sambor) (UA)	Fruits and bark used as a dye in Turkey [55].
*Vinca minor* L.	barwinek	sh	1	Easter eggs	green	SW Ukraine	
*Viola* sp. (probably *V. odorata* L.)	fiołki	f	2	Easter eggs	violet, blue	Rymanów (Krosno), Grąziowa (Ustrzyki Dolne)	Flowers are used as an amateur dye [56].
*Silene latifolia* ssp. *alba* (Mill.) Greuter and Burdet	sabaczeje mydło	n. sp.	1	washing clothes		NW Ukraine	
unidentified, either *Isatis tinctoria* L. or *Indigofera tinctoria* L.	indygo, indyk	n.sp.	3	textile, maybe also Easter eggs	n. sp.	Janów Lub., Maciejowice near Garwolin, Rabka	Imported; both widely used dye species.
unidentified	jabłonnik [not apple]	n. sp.		Easter eggs?	n. sp.		
unidentified	kacanki (pierwiosnek)	n. sp.		textiles?	n. sp.		
unidentified (maybe *Berberis vulgaris* L. though *Malus domestica* may not be excluded)	kwaśnica	bk	1	textiles	yellow	Mikulińce, Tarnopol, Zbaraż, Skałat, Nowesioło (UA)	
unidentified	leśnica	bark		textiles?	yellow		Probably hazel, which dyes yellow with alum (A.B.’s experiments).
unidentified	mieszalnik	n. sp.		Easter eggs?	n. sp.	Nieśwież, Słuck, Mińsk (BE)	Described as “forest grass”.
unidentified	mydło	n. sp.		textiles?	n. sp.	Tyłowo near Puck	Maybe a saponin-rich plant used as an element in preparing fabrics for dyeing. On the other hand, “gapie mydło” suggests *Lithospermum arvense*.
unidentified	pietruszka wodna	n. sp.		Easter eggs?	n. sp.	Łuków area	
unidentified	popawka	n. sp.		Easter eggs?	n. sp.	Lipów near Rzeczyca (BE)	
unidentified	wiluk	n. sp.		Easter eggs?	n. sp.	Lipów near Rzeczyca (BE)	
unidentified	zanogięć, zanagięć	n. sp.		Easter eggs?	yellow	Niedźwiadka near Łuków	
unidentified	zanowica	n. sp.		Easter eggs?	yellow, brown	Brus near Włodawa	
unidentified (could be *Lycopodium* sp.)	zielonka	n. sp.		Easter eggs?	n. sp.	Lipów near Rzeczyca (BE)	Described as “grass in conifer woods with branches similar to cypress, bright-green”. Also mentioned in *Gospodyni Litewska* [31].

bd—buds, bk—bark; br—branches; c—cones; fl—flowers, fr –fruits; l—leaves, r—roots; sh—young shoots; n. sp.—not specified; wd—wood; wh—whole plant with roots. More details: ^1^—After soaking textiles overnight, they become brown. They turn black if left in an iron-rich meadow soil. ^2^—One of the respondents said: “The old woman says that her parents dyed their wool and yarn black with buckthorn, but this is probably a mistake, for unripe buckthorn berries give a permanent yellow, ripe dark green”. ^3^—“in blue—they painted with commercial blue paint, but because this paint faded from the sun and was washed out by the rain, the housewives came up with an experiment to fix the color by soaking the yarn after dyeing in urine. For this purpose, a large bowl with two handles was ordered by the stove fitter, carefully stored from year to year in every peasant house for known use. All the elders participated in this activity, because a lot of fresh liquid was needed, and children were excluded, because if one of them said ‘it stinks’ the color would wear off. In general, dyeing activities were kept secret and not revealed to profane eyes”. ^4^—Podbielkowski [57] lists many synonyms for the species, e.g.,: “drzewo brazylijskie”, “drzewo fernambukowe”, “drzewo pernambukowe” (for *C. bras* and *C. echinata*). A related species, called sappanwood or eastern brazilwood (*Biancaea sappan* (L.) Tod.), is also used as a textile colorant to this day. Wood, bark, and roots are used, usually heartwood. The dye gives shades of red (orange, pink), depending on the mordants used and the pH of the dye bath (according to A.B.’s experience and [58]). In the 19th century, the names “drzewo fernambukowe” and “brazylka” were used [54]. We also suspect that in some sources from the 19th century, “brazylia” was used for *Haematoxylum campechianum* L., which gives blue and purple colors, e.g., according to [54] “Dyeing Easter eggs (…) red: decoction of ‘fernambuk’ with alum. For blue: decoction of ‘brazylia’”.

## Data Availability

The original data matrix is enclosed as Appendix A with the paper.

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
