# Peer review of "Plants and Other Materials Used for Dyeing in the Present Territory of Poland, Belarus and Ukraine according to Rostafiński’s Questionnaire from 1883"

_plants, 2023, doi:10.3390/plants12071482_

Round 1

Reviewer 1 Report

This is a very interesting paper that gathers a lot of information about ethnobotany and past uses of plants. A more detailed description of the context of the use of dyes could ameliorate the paper a lot. Eg readers would like to know more about the Polish cochineal or the Easter eggs. Coloured Easter eggs could be eaten or they were just decorative? Different social classes used different plants or colours, etc. Moreover I wonder why mushrooms are missing from the catalogue of natural dyes. If they do not exist in the data collected authors should make a comment about this. Lastly it would be nice in the conclusion of the authors to mention how this work could be practically helpful today. The phrase " the tradition of plant dyeing in Poland was already disappearing in the 19th c." is aphoristic. Certainly the practice is not common any more, but there is a trend of revival of such practices, so I do not believe that in Poland such initiatives do not exist today and they have completely disappeared. 

Concerning table 1:

1. Check letter size and style in all columns.

2 Titles in the first lines should start with a capital letter (Use, Main color etc).

3 Safflower: The latin name is missing

4 Haematoxylum: last column, "Turns" to be replaced with turns

5. If you use hay (with no reference to species) then you should do the same in the case of human urine or yolk.

6. Hypericum: you should explain how vodka is used as in the case of Hepatica or Vaccinium

7. Lichens: Any reference to characteristics or the biotope should be very useful.

8. Potentilla: Why are painting walking sticks?

9. Juglans regia: Easter eggs is correct in the column Use?

Author Response

This is a very interesting paper that gathers a lot of information about ethnobotany and past uses of plants. A more detailed description of the context of the use of dyes could ameliorate the paper a lot. Eg readers would like to know more about the Polish cochineal or the Easter eggs.

>we added more info on cochineal in intro and discussion about eggs in discussion section

Coloured Easter eggs could be eaten or they were just decorative?

>both, depending

Different social classes used different plants or colours, etc.

>this was a folk custom

 Moreover I wonder why mushrooms are missing from the catalogue of natural dyes. If they do not exist in the data collected authors should make a comment about this.

>no mushrooms were ever used, we added a comment on it.

Lastly it would be nice in the conclusion of the authors to mention how this work could be practically helpful today.

>We extended the conclusion.

The phrase " the tradition of plant dyeing in Poland was already disappearing in the 19th c." is aphoristic.

>well, we could not find a better phrase…. We are aware it is a bit banal.

Certainly the practice is not common any more, but there is a trend of revival of such practices, so I do not believe that in Poland such initiatives do not exist today and they have completely disappeared. 

>the second author of the paper is the author of the book on reviving traditional dyes and is active in teaching people to do so, this is mentioned in the bio part.

>We already mentioned it in the introduction: “Researching plant dyes is important not only from the point of view of recording tradi-tional knowledge. The information is also of value to modern enthusiasts of plant dyes. In the last two decades, there has been an immensely increasing trend in the use of natural products such as wild vegetables [42-43], medicinal herbs and teas [44], but al-so plant dyes. Several handbooks of traditional dyeing have been published recently [45-48].”

Concerning table 1:

  1. Check letter size and style in all columns.

>we did, we used a template after all.

2 Titles in the first lines should start with a capital letter (Use, Main color etc).

>done

3 Safflower: The latin name is missing

>Thank You for noticing! we corrected it.

4 Haematoxylum: last column, "Turns" to be replaced with turns

>strange… in the original file it is ‘turns’

  1. If you use hay (with no reference to species) then you should do the same in the case of human urine or yolk.

>no, because hay is multispecies and urine or yolk can be identified to species level

  1. Hypericum: you should explain how vodka is used as in the case of Hepatica or Vaccinium

for Hepatica and Vaccinum it was used as a dye, in case of Hypericum the plant is used only to give color to vodka. In case of Hepatica we modified to: the dye was made by extraction of flowers with vodka

  1. Lichens: Any reference to characteristics or the biotope should be very useful.

>we do not have more info than this provided

  1. Potentilla: Why are painting walking sticks?

> for decoration, walking stick were painted ree d – it is simple

  1. Juglans regia: Easter eggs is correct in the column Use?

>Yes, we only recorded it for eggs. In the 19th century the cultivation of walnuts was not common in Poland so we only have this reference.

Reviewer 2 Report

The manuscript is very interesting, not least because it collects some data not yet presented, although many references are already in the literature, but this can only corroborate them.

Author Response

As there were no comments we do not respond to this part.

Reviewer 3 Report

The paper reports the analysis of the questionnaires that Prof. Rostafiński used to study the dying plant in some Polish regions.

Even if the topic is of interest, some points must be addresses:

1. The paper is poor and weak in the section Results, that is consituted only by Table

2. The paragraph Discussion is poor without the punctual discussion that one would expect from a scientific journal dedicated to the study of plants

3. Conclusions should be re-written.

4. Table 1. Identification of species: It seems that the Authors of the present study identified the plants reported in Table 1. However, in Materials and Methods they declared that the species identification was made by the "standard methods applied in historical ethnobotany". Authors must explain these methods.

line 14: Abstract. Authors stated that the plants were "were identified by the respondents using Latin names". This is questionable as in another part Authors declared that onsly SOME repondents used latin binomials.

Lines 89-92: This sentence is unclear

Lines 96-100: This sentence is useless; please delete

Please chekc the binomials; for example Carthamus tinctorius in Table should be in italics.

Author Response

The paper reports the analysis of the questionnaires that Prof. Rostafiński used to study the dying plant in some Polish regions.

Even if the topic is of interest, some points must be addresses:

  1. The paper is poor and weak in the section Results, that is consituted only by Table

>we extended results and moved the long table to the end of the paper to make it more readable.

  1. The paragraph Discussion is poor without the punctual discussion that one would expect from a scientific journal dedicated to the study of plants

>we extended discussion

  1. Conclusions should be re-written.

>we extended conclusions

  1. Table 1. Identification of species: It seems that the Authors of the present study identified the plants reported in Table 1. However, in Materials and Methods they declared that the species identification was made by the "standard methods applied in historical ethnobotany". Authors must explain these methods.

>This was further developed in the same paragraph in the methods but we clarified it by adding: “i.e. comparing available voucher specimens, folk names recored in other sources and geographical distribution and abundance of the taxa used”

line 14: Abstract. Authors stated that the plants were "were identified by the respondents using Latin names". This is questionable as in another part Authors declared that onsly SOME repondents used latin binomials.

>Thank you for noticing this inaccuracy. We changed it to: They were identified by the respondents using folk names or sometimes even Latin names.

Lines 89-92: This sentence is unclear.

>We added explanation: “, as between 1795 and 1918 the independent Poland did not exist and was partitioned between these three empires”

Lines 96-100: This sentence is useless; please delete.

> We left it as Reviwer 1 even requested expanding this issue.

Please chekc the binomials; for example Carthamus tinctorius in Table should be in italics.

>corrected!

Round 2

Reviewer 3 Report

Even if the Authors have addressed the majority of my doubts, the focal point to be clarified is realted to the identification od plants reported in Table 1. The methods reported in the answer are questionable and vague. Please, re-write this part, responding to the question.

Author Response

We used similar description of our methods as in previous historical ethnobotanical publications, also those in ccoperation with other historical ethnobotany specialists, e.g. Ingvar Svanberg and Renata Soukand. Now we added a few of more sentences explaining our approach but this is generally clear for anyone dealing with historical ethnobotany. Please bear in mind that we took a conservative approach in the identification: in the case of thirteen plant names trustworthy identification was impossible. We provided more references about historical ethnobotany plant identification. We also had a large data set of dozens of other ethnobotanical studies from Poland with their plant names recorded.

Round 3

Reviewer 3 Report

No comment

Author Response

We disagree with the reviewer no. 3. We think that the methodology was explained in detail and the relevant references were cited. Similar methodology has been used in other historical ethnobotany publications in the same special issue “Historical Ethnobotany: Interpreting the Old Records”, e.g.
1.    Sõukand, R.; Kalle, R., 2022. The Appeal of Ethnobotanical Folklore Records: Medicinal Plant Use in Setomaa, Räpina and Vastseliina Parishes, Estonia (1888–1996). Plants 2022,11(20),2698.
2.    Dal Cero, M.,;Saller, R.; Leonti, M.; Weckerle, C.S. Trends of Medicinal Plant Use over the Last 2000 Years in Central Europe. ). Plants 2023, 12(1), p.135.
3.    Prakofjewa, J.; Anegg, M.; Kalle, R.; Simanova, A.; Prūse, B.; Pieroni, A.; Sõukand, R. Diverse in Local, Overlapping in Official Medical Botany: Critical Analysis of Medicinal Plant Records from the Historic Regions of Livonia and Courland in Northeast Europe, 1829–1895. Plants, 2022, 11(8), 1065.

We added these paper to the cited list of papers.